

# Variational truncated Wigner approximation for weakly interacting Bose fields: Dynamics of coupled condensates

C. D. Mink, A. Pelster, J. Benary, H. Ott and M. Fleischhauer⋆

Department of Physics and Research Center OPTIMAS, University of Kaiserslautern,
67663 Kaiserslautern, Germany

⋆ mfleisch@physik.uni-kl.de

## Abstract

The truncated Wigner approximation is an established approach that describes the dynamics of weakly interacting Bose gases beyond the mean-field level. Although it allows a quantum field to be expressed by a stochastic c-number field, the simulation of the time evolution is still very demanding for most applications. Here, we develop a numerically inexpensive scheme by approximating the c-number field with a variational ansatz. The dynamics of the ansatz function is described by a tractable set of coupled ordinary stochastic differential equations for the respective variational parameters. We investigate the non-equilibrium dynamics of a three-dimensional Bose gas in a one-dimensional optical lattice with a transverse isotropic harmonic confinement. The accuracy and computational inexpensiveness of our method are demonstrated by comparing its predictions to experimental data.



# 1   Introduction

In order to accurately describe the non-equilibrium dynamics of Bose condensates of atoms it is often necessary to treat the weakly interacting gas beyond the mean-field level [1], in particular if modes with low occupation become relevant. Prime examples, where such modes play an important role, are driven dissipative Bose gases with multi-stable stationary states or the filling dynamics of coupled condensates [2–9]. An approach, widely used in quantum optics, that takes both thermal and leading-order quantum fluctuations into account, is the truncated Wigner approximation (TWA) [10–18]. It amounts to approximating the boson quantum field by a classical field (c-field), which evolves in time according to the Gross-Pitaevskii equation. Although quantum fluctuations cannot be included exactly, they can be well approximated by sampling the initial value of the field over the Wigner quasi-distribution of the initial quantum state. In the presence of reservoir couplings additional noise sources have to be included leading to stochastic equations of motion [15, 19–22]. Since, in the initial-value sampling of the Wigner distribution, every unoccupied mode has fluctuations equivalent to one half of a particle, it is necessary to truncate the number of modes included, e.g. by appropriate projection techniques [23, 24]. For most applications the time evolution is, however, still intractable and prone to sampling noise.

In the present paper we develop a variational approach to the dynamics of the functional Wigner distribution, which we refer to as the variational truncated Wigner approximation (VTWA) for bosonic fields. To this end we decompose the classical field into a variational ansatz function and the remaining contribution, where we call the latter the residual field, which we here assume to remain in the vacuum state. In this way the functional Fokker-Planck equation obtained after performing the truncated Wigner approximation can be approximated by a finite-dimensional Fokker-Planck equation for the variational parameters. This allows the dynamics to be expressed by a tractable set of coupled stochastic differential equations. We apply our method to the non-equilibrium dynamics of a three-dimensional Bose gas in a one-dimensional optical lattice with a transversal isotropic harmonic confinement, where a single site is initially emptied [4]. We compare our predictions to experimental data and thereby demonstrate the accuracy and computational inexpensiveness of the VTWA.

In the paper we proceed as follows. In Sec. 2 we introduce the system of coupled condensates, which is a paradigmatic example of weakly interacting Bose fields, as its description requires a beyond mean-field treatment. This is followed by Sec. 3, which provides a concise general introduction to the TWA method. Subsequently, we develop in Sec. 4 a variational approach to the TWA, which represents the main result of our paper. This formalism is then applied to the dynamics of the coupled condensates and compared to experimental data in Sec. 5. Finally, Sec. 6 summarizes the results and gives an outlook.

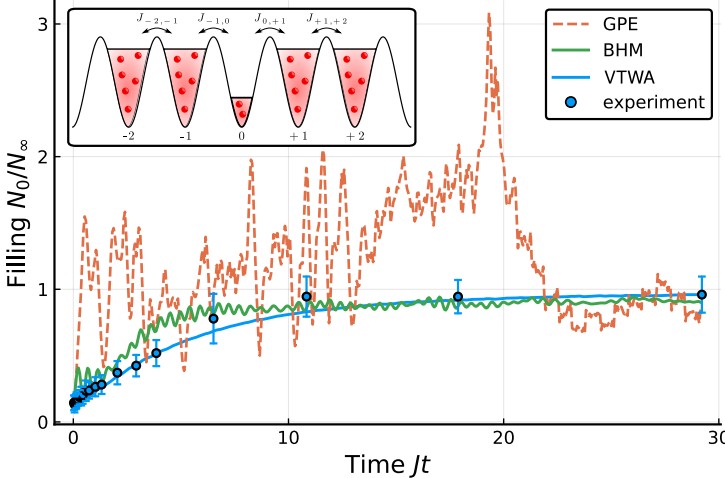

Figure 1: Refilling dynamics of the particle number $N_0$ of an initially emptied site in a one-dimensional optical lattice at a lattice depth of $8E_r$ obtained from the experiment as well as predicted by different theoretical approaches. The system consists of 31 sites with sites $i \neq 0$ starting with $N_\infty = 940$ particles and has open boundary conditions. For the Bose-Hubbard model $J/U = 40$ is chosen to fit the experiment, whereas the GPE and VTWA assume the Hamiltonian of eq. (1) and parameters given in Sec. 5.1, i.e. do not use any fitting parameters. Insert: Schematic representation of the experiment. A particle current towards the central lattice site is induced by the initial out-of-equilibrium configuration. The overlap of nearest-neighbor wave functions leads to dynamical tunneling coefficients $J_{ij}$.

## 2 Coupled condensates

We start with illustrating the variational approach and consider as an example the dynamics of weakly interacting Bose condensates in situations, where all except very few relevant modes have a macroscopic occupation of particles. To this end we study a weakly interacting Bose gas in an optical lattice with lattice period $a$ along the $z$-direction and an isotropic harmonic confinement of frequency $\omega_r$ in the $x, y$-direction. The many-body Hamiltonian of this system in units of length $a$ and the recoil energy of the lattice $E_r = (\hbar\pi)^2/2ma^2$ reads

$$\hat{H} = \int d^3r \, \hat{\psi}^\dagger(\vec{r})\left[\mathcal{H}_0 + \frac{U_0}{2}\hat{\psi}^\dagger(\vec{r})\hat{\psi}(\vec{r})\right]\hat{\psi}(\vec{r}), \tag{1}$$

where the single-particle Hamiltonian is given by

$$\mathcal{H}_0 = -K\Delta + V_z \sin^2(\pi z) + V_r(x^2 + y^2). \tag{2}$$

Here $\hat{\psi}(\vec{r})$, $\hat{\psi}^\dagger(\vec{r})$ are bosonic field operators with the canonical commutation relation $[\hat{\psi}(\vec{r}), \hat{\psi}^\dagger(\vec{s})] = \delta(\vec{r}-\vec{s})$ and $\Delta = \vec{\nabla}\cdot\vec{\nabla}$ is the Laplace operator. Furthermore, $V_z$ denotes the lattice depth in units of the recoil energy and we have $K \equiv \pi^{-2}$, $V_r \equiv m\omega_r^2 a^2/2E_r$, $U_0 \equiv g/a^3 E_r$, with $m$ being the boson mass and $g$ the interaction strength in s-wave scattering approximation.

The commonly applied Gross-Pitaevski mean-field equation (GPE) is not sufficient to describe the dynamics of coupled BECs with a single initially empty site, a situation where the quantum fluctuations become important. This can clearly be seen in Fig. 1, where the refilling

dynamics of an initially emptied site is obtained with a variational ansatz for the GPE wave function according to Appendix A. It shows strong oscillations of the particle number as well as abrupt jumps. Both of these traits are not observed experimentally and are an artifact of the lack of quantum fluctuations in the mean-field description. Thus, more accurate theoretical predictions require taking leading-order quantum fluctuations into account. A full quantum treatment is possible, on the other hand, by neglecting the three-dimensional character of the Bose gas. Then one can approximate the coupled condensates by a one-dimensional Bose-Hubbard model (BHM) given by the Hamiltonian

$$\hat{H}_{\text{BHM}} = \hbar\omega \sum_j \hat{a}_j^\dagger \hat{a}_j + \frac{\hbar U}{2} \sum_j \hat{a}_j^\dagger \hat{a}_j^\dagger \hat{a}_j \hat{a}_j - \hbar J \sum_{\langle jk \rangle} \hat{a}_j^\dagger \hat{a}_k \,, \tag{3}$$

where $\hat{a}_j, \hat{a}_j^\dagger$ are bosonic creation and annihilation operators at the $j$th lattice site, satisfying $[\hat{a}_j, \hat{a}_k^\dagger] = \delta_{jk}$, and $\langle jk \rangle$ denotes summation over nearest neighbors. The reduction to a one-dimensional model allows a full quantum mechanical description using e.g. tDMRG (time-dependent Density Matrix Renormalization Group) or other numerical techniques based on matrix-product expansions of the many-body state [25]. Ignoring the transverse degrees of freedom in this effective model does, however, neglect that the tunneling between adjacent sites depends on the number of atoms in these sites. Choosing adapted values for the effective tunneling matrix element $J$ and the interaction strength $U$ the BHM predicts a smoother re-filling process as compared to the GPE with overall smaller oscillations, which are still strong in the initial phase. Since the refilling process is faster and the initial oscillations are not detected experimentally, previous attempts of modeling the dynamics using the BHM and similar effective-mode models postulated the presence of an additional dephasing process [26,27].

In contrast to both the mean-field GPE description and the reduction to an effective BHM, our variational truncated Wigner method, which is worked out below, predicts negligible oscillations without requiring any additional couplings to reservoirs. Furthermore, the small initial transversal overlap of neighboring Bose gas wave functions suppresses the tunneling, yielding an accurate match with the experiment without using any fitting parameters.

## 3 Truncated Wigner approximation for weakly interacting bosons

A common approach to describe weakly interacting bosons in the limit of large mode occupation is the truncated Wigner approximation (TWA) [10, 13, 14, 17, 18]. In the following we summarize the key ingredients of this approach, first for a single mode and then for a continuum Bose field.

### 3.1 Wigner phase-space representation

The TWA is based on the Wigner distribution of a bosonic mode, $\hat{a}, \hat{a}^\dagger$, which is obtained by expanding the density operator $\hat{\rho}$ in the overcomplete set of coherent states $|\alpha\rangle$ with $\alpha \in \mathbb{C}$ [28]:

$$W(\alpha, \alpha^*) = \int \frac{d^2\eta}{2\pi^2} \langle \alpha - \frac{\eta}{2} | \hat{\rho} | \alpha + \frac{\eta}{2} \rangle e^{(\eta^* \alpha - \eta \alpha^*)/2} \,. \tag{4}$$

The expectation value of symmetrically ordered operator products is then given by an average over the Wigner distribution:

$$\langle (\hat{a}^{\dagger m} \hat{a}^n)_{\text{symm}} \rangle = \int d^2\alpha \, W(\alpha, \alpha^*) (\alpha^*)^m \alpha^n \,. \tag{5}$$

The effect of a ladder operator acting on the density operator can be equivalently expressed as a differential identity for the Wigner distribution, where the extension to many modes is straightforward [29]:

$$
\begin{aligned}
\hat{a}_j \hat{\rho} &\leftrightarrow \left(\alpha_j + \tfrac{1}{2}\tfrac{\partial}{\partial \alpha_j^*}\right)W, \quad \hat{a}_j^\dagger \hat{\rho} \leftrightarrow \left(\alpha_j^* - \tfrac{1}{2}\tfrac{\partial}{\partial \alpha_j}\right)W, \\
\hat{\rho}\hat{a}_j &\leftrightarrow \left(\alpha_j - \tfrac{1}{2}\tfrac{\partial}{\partial \alpha_j^*}\right)W, \quad \hat{\rho}\hat{a}_j^\dagger \leftrightarrow \left(\alpha_j^* + \tfrac{1}{2}\tfrac{\partial}{\partial \alpha_j}\right)W.
\end{aligned}
\tag{6}
$$

Here $W\left(\{\alpha_i, \alpha_i^*\}\right) \in \mathbb{R}$ denotes the Wigner distribution that associates a quasi-probability with every point $\vec{p} = (\alpha_1, \alpha_1^*, \alpha_2, \alpha_2^*, \dots)^T$ in the coherent state phase space [30, 31]. Note that $\alpha_i, \alpha_i^*$ are treated as independent variables. Applying these mappings to the Von-Neumann equation of the density operator $\hat{\rho}$ yields a partial differential equation for the propagation of the Wigner distribution. For the BHM of Eq. (3) the corresponding equation reads

$$
\begin{aligned}
\frac{\partial}{\partial t}W = &-i \sum_j \frac{\partial}{\partial \alpha_j}\left\{\left[-\omega - U\left(|\alpha_j|^2 - \frac{1}{2}\right)\right]\alpha_j + J(\alpha_{j-1} + \alpha_{j+1})\right\}W \\
&+ \frac{iU}{4}\sum_j \frac{\partial^3}{\partial \alpha_j^2 \partial \alpha_j^*}\alpha_j W + \text{c.c.},
\end{aligned}
\tag{7}
$$

where the time $t$ is measured in units of $t_r = \hbar/E_r$. This equation is in general difficult to solve. One notices, however, that the terms with third derivatives scale with the inverse occupation $|\alpha_j|^{-2} \approx \langle \hat{a}_j^\dagger \hat{a}_j \rangle^{-1}$ of each mode. A powerful approximation, justified in the case of weak interaction $U$ and macroscopic occupation of all modes, consists of neglecting these third order terms and is called the truncated Wigner approximation (TWA). The equation remaining after truncating higher than second order derivatives is a multivariate Fokker-Planck equation (FPE). Every FPE has an equivalent set of stochastic differential equations (SDEs) for complex-valued stochastic functions $\alpha_j(t)$. In many cases, such as the unitary BHM, the diffusion matrix vanishes exactly and the SDEs are reduced further to nonlinear ordinary differential equations:

$$
\dot{\alpha}_j = -i\left(\tilde{\omega} + U|\alpha_j|^2\right)\alpha_j + iJ(\alpha_{j-1} + \alpha_{j+1}),
\tag{8}
$$

where $\tilde{\omega} = \omega - U/2$. The initial values of $\alpha_j(t_0) = \alpha_{j0}$ are non-deterministic and are represented by a given initial Wigner distribution. If the Wigner distribution happens to be positive semi-definite, one can sample initial values $\alpha_{j0}$ in the coherent state phase space and obtain their subsequent time evolution by integrating the SDEs such as Eq. (8). Finally, expectation values are calculated by evaluating operators in their symmetrical order:

$$
\langle (\hat{a}^{\dagger m}\hat{a}^n)_{\text{symm}} \rangle = \overline{(\alpha^*)^m \alpha^n}.
\tag{9}
$$

The bar indicates the average with respect to the many numerically time-evolved trajectories. Since it can be rather tedious to calculate the symmetrical order of a given function $f(\hat{a}^\dagger, \hat{a})$ by manually applying the commutation relationship, it is useful to introduce the Bopp operators [32]:

$$
\hat{a} = \alpha + \frac{1}{2}\frac{\partial}{\partial \alpha^*}, \qquad \hat{a}^\dagger = \alpha^* - \frac{1}{2}\frac{\partial}{\partial \alpha},
\tag{10}
$$

which translates products of operators into their corresponding c-numbers by letting them act on the scalar 1. Consider the particle number:

$$
\hat{a}^\dagger \hat{a} = \left(\alpha^* - \frac{1}{2}\frac{\partial}{\partial \alpha}\right)\left(\alpha + \frac{1}{2}\frac{\partial}{\partial \alpha^*}\right)1 = |\alpha|^2 - \frac{1}{2}.
\tag{11}
$$

Hence we obtain $\langle(\hat{a}^\dagger\hat{a})_{\text{symm}}\rangle = \overline{|\alpha|^2} - 1/2$. An important consequence of this symmetric averaging is that the vacuum $|0\rangle\langle0|$ has a virtual occupation of half a particle, which is subtracted *after* the averaging is performed. This yields the correct result of zero particles in the vacuum mode.

At first glance, and apart from the irrelevant term $U/2$ which is absorbed into the definition of $\tilde{\omega}$, Eq. (8) is identical to the Gross-Pitaevskii-like mean-field equation of the Bose-Hubbard model. However, in order to evaluate symmetrically ordered expectation values one still has to average the solution of Eq. (8) over stochastic initial conditions. Therefore, the TWA goes in fact beyond the mean-field approximation and also takes lowest-order quantum fluctuations into account. As illustrated in Appendix B, it turns out to be equivalent to the Bogoliubov approximation in case of the Bose-Hubbard model. Note that further quantum corrections to the dynamics of interacting bosons beyond the truncated Wigner approximation have also been worked out [14].

## 3.2 Wigner functional of quantum fields

Now consider the continuum description of a Bose gas in terms of bosonic field operators $\hat{\psi}(\vec{r}), \hat{\psi}^\dagger(\vec{r})$. There are two different formulations of the Wigner representation for such a field, which are relevant for our discussion.

First, for a given set of orthonormal functions $V = \{\chi_n(\vec{r})\,|\,n \in \mathbb{N}\}$, e.g. plane waves or harmonic oscillator eigenfunctions, a decomposition into discrete modes and corresponding coherent amplitudes can be defined

$$\hat{\psi}(\vec{r}) = \sum_{n=0}^{N} \chi_n(\vec{r})\hat{a}_n \longleftrightarrow \psi(\vec{r}) = \sum_{n=0}^{N} \chi_n(\vec{r})\alpha_n \,, \tag{12}$$

where for all practical purposes $N < \infty$, i.e. a truncation to a finite amount of single-particle low-energy eigenstates, must be chosen. By inserting this expansion into a given Hamiltonian or Lindbladian, we can derive equations of motion for the coherent amplitudes corresponding to the discrete ladder operators. However, for sufficiently large $N$, modes with non-macroscopic occupation occur and the TWA is typically not justified anymore. One possibility of remedying this for the case of harmonic confinement is contained in the formalism of the *stochastic projected Gross-Pitaevskii Equation* [24].

Secondly, when taking the limit $N \to \infty$, it is possible to derive mappings between the field operators and functional derivatives of a quasi-probability distribution as was shown, for instance, in case of the Glauber-Sudarshan P-distribution [29]. For the Wigner representation one obtains analogously

$$\hat{\psi}(\vec{r})\hat{\rho} \longleftrightarrow \left[\psi(\vec{r}) + \frac{1}{2}\frac{\delta}{\delta\psi^*(\vec{r})}\right]W\,, \tag{13a}$$

$$\hat{\psi}^\dagger(\vec{r})\hat{\rho} \longleftrightarrow \left[\psi^*(\vec{r}) - \frac{1}{2}\frac{\delta}{\delta\psi(\vec{r})}\right]W\,, \tag{13b}$$

$$\hat{\rho}\hat{\psi}(\vec{r}) \longleftrightarrow \left[\psi(\vec{r}) - \frac{1}{2}\frac{\delta}{\delta\psi^*(\vec{r})}\right]W\,, \tag{13c}$$

$$\hat{\rho}\hat{\psi}^\dagger(\vec{r}) \longleftrightarrow \left[\psi^*(\vec{r}) + \frac{1}{2}\frac{\delta}{\delta\psi(\vec{r})}\right]W\,, \tag{13d}$$

with the Wigner functional $W[\psi(\vec{r}), \psi^*(\vec{r})]$, where $\psi(\vec{r}), \psi^*(\vec{r})$ are treated as independent fields. Similarly, we can also derive Bopp operators by composition of the single-mode expressions of Eq. (10):

$$\hat{\psi}(\vec{r}) = \psi(\vec{r}) + \frac{1}{2}\frac{\delta}{\delta\psi^*(\vec{r})}, \quad \hat{\psi}^\dagger(\vec{r}) = \psi(\vec{r}) - \frac{1}{2}\frac{\delta}{\delta\psi(\vec{r})}\,. \tag{14}$$

It is important to realize that the integral over the density $|\psi(\vec{r})|^2$ diverges since, according to Eq. (12), even the vacuum has on average $\lim_{N\to\infty} N/2$ particles before subtracting the symmetrical ordering correction. Combining the mappings in Eqs. (13) with the TWA, which is now justified in regions $\vec{r} \in R$ where macroscopic occupation occurs, it is possible to obtain a functional FPE.

## 4 Dynamical variational reduction of the functional Fokker-Planck equation

While the complexity of the dynamics of the quantum field has already been reduced by introducing a truncated Wigner phase space description, integrating the functional FPE or the corresponding stochastic partial differential equation is still numerically challenging if not unfeasible. Let us therefore aim for a natural stochastic extension of the variational approximation of the deterministic GPE dynamics, which was pioneered in [33]. To this end we consider the case where the region $R \subset \mathbb{R}^n$ of the field $\psi(\vec{r})$ can be approximated by a variational ansatz $\psi^{(0)}(\vec{c}, \vec{r})$ with $\vec{c} \in \mathbb{R}^M$ denoting a suitable set of $M$ variational parameters and let $\psi^{(1)}(\vec{c}, \vec{r}) \equiv \psi(\vec{r}) - \psi^{(0)}(\vec{c}, \vec{r})$ be the residual field of low density. Our goal is then to transform the functional FPE onto a multivariate FPE in $\vec{c}$ as outlined in detail in Appendix C. Since explicit expressions for the functionals $\vec{c}[\psi, \psi^*]$ and $\psi^{(0)}[\psi, \psi^*]$ are generally not obtainable, the functional derivatives required (see Eqs. (50)) cannot be obtained from these. However, since only the functional derivatives of the new variables are needed, we can calculate them through implicit differentiation. To this end we define the functional

$$F = \int d\vec{r} \, |\psi(\vec{r}) - \psi^{(0)}(\vec{c}, \vec{r}) - \psi^{(1)}(\vec{c}, \vec{r})|^2 \,, \tag{15}$$

which has the global minimum $F = 0$ when $\psi = \psi^{(0)} + \psi^{(1)}$. The extremalization

$$F_{c_i} = \frac{\partial F}{\partial c_i} = 0 \,, \tag{16a}$$

$$F_{\psi(\vec{r})} = \frac{\delta F}{\delta \psi^{(1)}(\vec{r})} = 0 \,, \tag{16b}$$

yields a set of equations that implicitly define the new variables. The functional derivatives of these equations with respect to the original field $\psi, \psi^*$ are connected to the desired derivatives via the chain rule:

$$0 = \frac{\delta F_{c_i}}{\delta \psi(\vec{r})} = -\frac{\partial}{\partial c_i}(\psi^{(0)} + \psi^{(1)})^* + \sum_j R_{ij} \frac{\delta c_j}{\delta \psi(\vec{r})} \,. \tag{17}$$

The coefficients $R_{ij} = \partial^2 F / \partial c_j \partial c_i$ define a $M \times M$ matrix. At the global minimum $F = 0$ we obtain:

$$R_{ij} = \int d\vec{r} \left[ \frac{\partial(\psi^{(0)} + \psi^{(1)})^*}{\partial c_i} \frac{\partial(\psi^{(0)} + \psi^{(1)})}{\partial c_j} + \text{c.c.} \right] \,. \tag{18}$$

Assuming that $R$ is invertible, we obtain from Eq. (17):

$$\frac{\delta c_i}{\delta \psi(\vec{r})} = \sum_j \left(R^{-1}\right)_{ij} \frac{\partial}{\partial c_j}(\psi^{(0)} + \psi^{(0)})^* \,. \tag{19}$$

Correspondingly, the second derivatives are obtained from functional derivatives of $F_{c_i}$. For example

$$
\frac{\delta^2 c_i}{\delta \psi^*(\vec{r})\delta \psi(\vec{s})} = \sum_j (R^{-1})_{ij} \int d\vec{r} \left[ \frac{\partial^2(\psi^{(0)}+\psi^{(1)})}{\partial c_i \partial c_j} \frac{\delta c_j}{\delta \psi^*(\vec{r})} + \text{c.c.} \right]
$$
$$
- \sum_{jkl} (R^{-1})_{ij} S_{jkl} \frac{\delta c_k}{\delta \psi^*(\vec{r})} \frac{\delta c_l}{\delta \psi(\vec{s})}, \tag{20}
$$

where

$$
S_{jkl} = \int d\vec{r} \left[ \frac{\partial(\psi^{(0)}+\psi^{(1)})^*}{\partial c_j} \frac{\partial^2(\psi^{(0)}+\psi^{(1)})}{\partial c_j \partial c_k} + \frac{\partial^2(\psi^{(0)}+\psi^{(1)})^*}{\partial c_j \partial c_l} \frac{\partial(\psi^{(0)}+\psi^{(1)})}{\partial c_k} \right.
$$
$$
\left. + \frac{\partial(\psi^{(0)}+\psi^{(1)})^*}{\partial c_l} \frac{\partial^2(\psi^{(0)}+\psi^{(1)})}{\partial c_j \partial c_k} + \text{c.c.} \right]. \tag{21}
$$

The derivatives of $\psi^{(1)}$ are similarly accessible. In Sec. 5.2 we assume that the coupling between the coefficients $\vec{c}$ and the residual field $\psi^{(1)}(\vec{r})$ is weak and can be neglected. In the case of an initial factorization $W(0) = W(\vec{c},0)W[(\psi^{(1)})^*,\psi^{(1)}](0)$, the Wigner function will always remain factorized and the residual field can be ignored. This yields a multivariate FPE in $\vec{c}$

$$
\frac{\partial}{\partial t} W(t,\vec{c}) = \mathcal{L}(\vec{c}) W(t,\vec{c}), \tag{22}
$$

whose coefficients are given by Eqs. (50). Note that (near-) vacuum fluctuations of the low-density region are neglected and only fluctuations that can be parameterized by the variational ansatz $\psi^{(0)}(\vec{c},\vec{r})$ remain. These are included in the dynamics by sampling the initial coefficients $\vec{c}(0)$ from the reduced Wigner distribution. The reduction to a finite set of variational parameters without including effects induced by $\psi^{(1)}(\vec{r})$ turns out to be a suitable approximation for the example of coupled condensates near absolute zero temperature discussed here.

While this approach greatly reduces the complexity of the time evolution, it is still not clear how initial states and expectation values in the functional limit can be obtained. We demonstrate these remaining problems by working out a concrete model.

# 5 Non-equilibrium dynamics of a weakly interacting Bose gas in a 1d optical lattice

To illustrate the formalism derived in the previous section and validate its results, we apply it to the optical lattice Hamiltonian of Eq. (1), describing tunnel-coupled Bose-Einstein condensates, and compare it to the experimental results.

## 5.1 Experimental setup

In the experimental realization a weakly interacting BEC of about $1.9 \cdot 10^5$ $^{87}$Rb atoms is created in a single beam dipole trap with frequencies $\omega_z/2\pi = 12\,\text{Hz}$ along the $z$-axis and $\omega_r/2\pi = 227\,\text{Hz}$ in the perpendicular directions. The cigar-shaped BEC is adiabatically loaded into a 1d optical lattice generated by two blue detuned laser beams crossing at an angle of $90°$. The resulting lattice has a period of $a = 547\,\text{nm}$ along the $z$-axis. At the center of the trap each site contains a small, pancake-shaped BEC with about $N_\infty = 940$ atoms and the total number of lattice sites is about 200. Using a well-defined particle loss originating from a

scanning electron microscopy technique (SEM) the central site of the lattice is emptied [34]. This is done within 5 ms at a lattice depth of $V_z = 30\,E_r$, where $E_r = (\hbar\pi)^2/2ma^2$ is the recoil energy of the lattice. Thereafter the lattice depth is ramped down to the desired value for the measurement run while keeping the dissipation process on. At the end of the preparation phase the dissipation is switched off. After variable times the lattice depth is set to a value of $V_z = 30\,E_r$ again, effectively freezing out the motion between lattice sites. An image of the density distribution in the optical lattice is then taken with the scanning electron microscope. From these images the relative filling of the central site and the distribution of atoms within this site is obtained. Inspired by the previous experimental results in [35], new experiments were conducted for this paper to study the refilling dynamics quantitatively. In particular, the goal of these new experiments was to carry out a systematic investigation how the refilling behaviour depends on the lattice depth as well as to gain first insights into how fluctuation effects vary as a function of time.

## 5.2 Variational TWA for coupled condensates

Where not otherwise noted, we will from now on assume the parameters given in Sec. 5.1. The lattice Hamiltonian eq. (1) describes a system that is translationally invariant in the $z$-direction and, thus, does not take into account the shallow harmonic trapping along the $z$-axis, present in the experiment. We neglect this confinement in the $z$-direction, which is valid at the center of the trap. Due to the low temperatures of the condensates, which are much smaller than their critical temperature, it is justified to consider the residual modes $\psi_1(\vec{r})$ to be vacuum states. Thus, we will restrict ourselves in the following to the mode $\psi_0(\vec{r})$. The functional mappings produce only first and third derivatives of which the latter can be neglected in TWA. Thus, we obtain a functional FPE with the drift coefficients:

$$D_{\psi(\vec{r})} = iK\Delta\psi - iV_r(x^2 + y^2)\psi - iU_0|\psi|^2\psi, \tag{23a}$$

$$D_{\psi^*(\vec{r})} = \left(D_{\psi(\vec{r})}\right)^*, \tag{23b}$$

and a vanishing diffusion matrix. We postulate that the relevant effects in this system are breathing motions at each site and particle exchanges between nearest-neighbor sites. These are represented by the ansatz:

$$\psi^{(0)}(\vec{r}) = \sum_j w_{j,0}(z)\psi_j(x, y), \tag{24}$$

$$\psi_j(x, y) = (\pi\sigma_j^2)^{-1/2}\sqrt{N_j}e^{-i\phi_j}$$
$$\cdot \exp\left[\left(-\frac{1}{2\sigma_j^2} + iA_j\right)(x^2 + y^2)\right], \tag{25}$$

where $w_{j,0}(z)$ is the lowest-band Wannier function of the single-particle Schrödinger equation at the $j$th potential minimum of the optical lattice. It satisfies the orthonormality relation:

$$\delta_{jk} = \int dz\, w_{j,0}^*(z)w_{k,0}(z). \tag{26}$$

The variational parameters occurring in Eq. (25) are referenced by a double index, where the Latin character refers to the site:

$$c_{j,\alpha} = (N_j, \phi_j, \sigma_j, A_j)_\alpha, \qquad \alpha = 1, \ldots, 4. \tag{27}$$

In the following steps we assume $\partial_{c_{j,\alpha}}\psi^{(0)} \gg \partial_{c_{j,\alpha}}\psi^{(1)}$, thus completely decoupling the dynamics of $\psi^{(0)}$ from the residual field $\psi^{(1)}$. The matrix $R$ of Eq. (18) is now a block diagonal matrix

with Latin indices referring to the blocks. The inverse is given by inverting each diagonal block individually:

$$(R^{-1})_{j\alpha,k\beta} = \frac{\delta_{jk}}{2N_j} \begin{pmatrix} 4N_j^2 & 0 & 0 & 0 \\ 0 & 2 & 0 & \sigma_j^{-2} \\ 0 & 0 & \sigma_j^2 & 0 \\ 0 & \sigma_j^{-2} & 0 & \sigma_j^{-4} \end{pmatrix}_{\alpha\beta}. \tag{28}$$

Furthermore, in order to shorten our notation, we define the overlap coefficients

$$U_{\text{eff}} = U_0 \int dz \, |w_{j,0}(z)|^4, \tag{29a}$$

$$J = \int dz \, w_{j,0}^*(z) \left[ K \frac{\partial^2}{\partial z^2} - V_z \sin^2(\pi z) \right] w_{j+1,0}(z), \tag{29b}$$

$$\eta_{jk}^{(n)} = \int dx\,dy \, (x^2 + y^2)^{n/2} \psi_j^*(x,y) \psi_k(x,y). \tag{29c}$$

Finally we only keep diagonal interaction contributions and nearest-neighbor tunneling terms. From the drift coefficients and vanishing diffusion coefficients we immediately obtain the following set of coupled ordinary differential equations for the variational parameters:

$$\dot{N}_j = -2J \sum_{k=j\pm 1} \text{Im}(\eta_{jk}^{(0)}), \tag{30a}$$

$$\dot{\phi}_j = \left( 2K + \frac{3U_{\text{eff}}}{4\pi} N_j \right) \sigma_j^{-2} - J N_j^{-1} \sum_{k=j\pm 1} \text{Re}\left( \eta_{jk}^{(0)} - \sigma_j^{-2} \eta_{jk}^{(2)} \right), \tag{30b}$$

$$\dot{\sigma}_j = 4K A_j \sigma_j + J(\sigma_j N_j)^{-1} \sum_{k=j\pm 1} \text{Im}\left( \sigma_j^2 \eta_{jk}^{(0)} - \eta_{jk}^{(2)} \right), \tag{30c}$$

$$\dot{A}_j = -V_r - 4K A_j^2 + \left( K + \frac{U_{\text{eff}}}{4\pi} N_j \right) \sigma_j^{-4} - J(\sigma_j^2 N_j)^{-1} \sum_{k=j\pm 1} \text{Re}\left( \eta_{jk}^{(0)} - \sigma_j^{-2} \eta_{jk}^{(2)} \right). \tag{30d}$$

Note that in Appendix A the same differential equations are obtained from restricting the deterministic GPE to the same variational ansatz. The change of the particle number $\dot{N}_j$ is proportional to $\sqrt{N_j N_{j\pm 1}}$ which is reminiscent to Josephson tunneling. However, a striking difference is the additional appearance of the transversal overlap in $\eta_{jk}^{(0)}$, see Eq. (29c), which acts as a dynamical weight to the tunneling rate $J$. Since the system is point-symmetric with respect to the initially empty site, we only integrate the positive lattice sites and mirror them to obtain the dynamics of the negative lattice sites. In our simulations, we evolve 16 sites, i.e. the central site and its 15 nearest-neighbors to the right. In conjunction with the point-symmetry this results in a total of 31 time-evolved sites. Open boundary conditions after the ±15 sites are chosen.

In order to generate an initial state of the coupled condensates we set both tunneling $J = 0$ and interaction $U_{\text{eff}} = 0$. The decoupled sites are 2d harmonic oscillators and we express their initial states in terms of the $m$th oscillator eigenfunction $\chi_{j,m}(x)$ of site $j$:

$$\psi_j(x,y,t=0) = \sum_{m,n=0}^{\infty} \chi_{j,m}(x) \chi_{j,n}(y) \alpha_{j,mn}, \tag{31}$$

where $\alpha_{j,mn}$ is a fluctuating coherent amplitude determined from the initial Wigner function. Furthermore, we identify the initial macroscopic region as

$$\psi_j^{(0)}(x,y,t=0) = \chi_{j,0}(x) \chi_{j,0}(y) \sqrt{N_j} e^{-i\phi_j}, \tag{32}$$

which can be expressed in terms of our variational ansatz with $\sigma_j$ being equal to the harmonic oscillator length $\sigma_{\mathrm{ni}} = (K/V_r)^{1/4}$ in optical lattice units and $A_j = 0$. The Wigner fluctuations therefore enter the dynamics via the non-deterministic initial conditions of $N_j$ and $\phi_j$. The remaining terms are identified as the residual field $\psi_j^{(1)}(x,y,t=0) = \sum_{m,n=1}^{\infty} \chi_{j,m}(x)\chi_{j,n}(y)\alpha_{j,mn}^{\mathrm{vac}}$ where the complex normally distributed $\alpha_{j,mn}^{\mathrm{vac}}$ represent the vacuum fluctuations of each mode. The ground state is obtained by assuming a Fock state [36] with $N_{\infty} = 940$ mean occupation in the ground state mode for the initially full sites and $\sim 15\% \, N_{\infty}$ for the initially empty site depending on the lattice depth. The interacting ground state is then generated by an adiabatic ramp up of the interaction strength $U_{\mathrm{eff}}$ up to a desired value. Then the tunneling coefficient $J$ is changed from 0 to the target value corresponding to the lattice depth $V_z$ thus initiating the refilling process of the central site.

For each lattice depth $3 \cdot 10^4$ trajectories are evolved. Using the *DifferentialEquations.jl* package [37] for the *Julia Programming Language* [38] the calculation of a single trajectory to 100 ms takes $\sim 1$ ms per thread on an *Intel Xeon Gold 6126* processor with a base clock frequency of 2.60 GHz.

### 5.2.1 Refilling dynamics

We first discuss the dynamics of the particle number at each potential minimum of the optical lattice. To obtain the mean occupation of a given site we average the trajectories according to:

$$\langle \hat{N}_j \rangle = \int d\vec{r} \left\{ \overline{|\psi_j(\vec{r})|^2} - \frac{1}{2}\delta(\vec{0}) \right\}$$
$$= \int dx\,dy\, \overline{|\psi_j^{(0)}(x,y)|^2} - \frac{1}{2} = \overline{N_j} - \frac{1}{2}. \tag{33}$$

The diverging Dirac delta function $\delta(\vec{0})$ is a consequence of the symmetrical ordering of the field operators. Since we assume $\psi_j^{(1)}(\vec{r})$ to be the vacuum in all modes orthogonal to $\psi_j^{(0)}(\vec{r})$, it cancels with the delta function, leaving only a single $1/2$ to account for the fluctuations in the mode $\psi_j^{(0)}(\vec{r})$. The radial width of the condensate and its fluctuations are given by:

$$\langle \hat{\sigma}_j^2 \rangle = \langle \hat{N}_j \rangle^{-1} \int dx\,dy\,(x^2+y^2) \overline{|\psi_j^{(0)}(x,y)|^2} = \langle \hat{N}_j \rangle^{-1} \overline{N_j \sigma_j^2}, \tag{34a}$$

$$(\Delta \sigma_j^2)^2 = \langle \hat{N}_j \rangle^{-2} \left( \overline{N_j^2 \sigma_j^4} - \overline{N_j \sigma^2}^2 \right). \tag{34b}$$

The mean occupation numbers and radial widths of the central site and its two-nearest neighbors, obtained from VTWA simulations, are depicted in Fig. 2. The radial width $\sigma_0(t)$ converges much faster to the value of its neighbors than the occupation $N_0(t)$. Furthermore, a breathing motion with frequency $\omega = 2\omega_r$ is induced [39]. The fluctuations of the transversal width $\Delta\sigma_0 = \sqrt{\Delta\sigma_0^2}$ are strongest while the width rapidly increases and then slowly decay. The mean occupation shows weak initial oscillations that decay during the refilling process.

In Fig. 3a) the time evolution of the number of atoms in the central site is shown for increasing lattice depths $V_z$ and compared to experimental data. While for small values $V_z$ the refilling curve has an exponential shape, an initial slowing down occurs in deeper lattices and a pronounced s-shape arises, which can be approximated by a logistic function

$$N(t) = \frac{N_{\infty}N_0}{N_{\infty}e^{-t/\tau} + N_0(1 - e^{-t/\tau})}. \tag{35}$$

One recognizes a rather good quantitative agreement between our theoretical model and the experimental data for the smaller lattice depths of $V_z = 6\,E_r$ and $V_z = 8\,E_r$. However, at

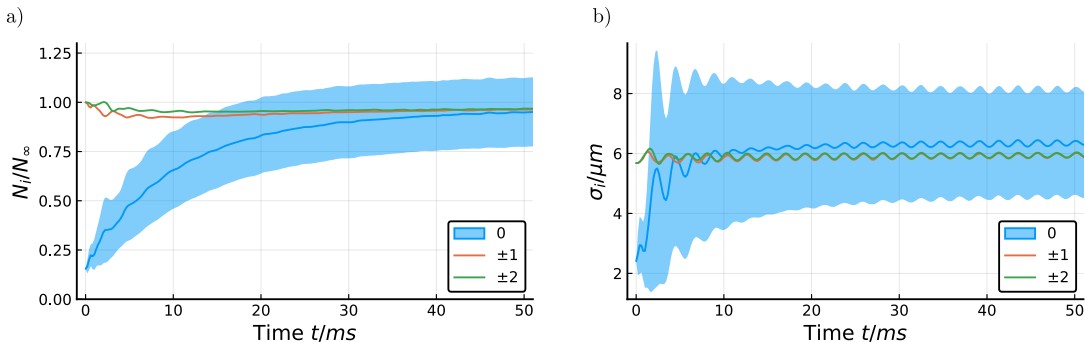

Figure 2: Dynamics of a) fillings and b) transversal widths of the central atomic cloud and its two nearest-neighbors at $V_{\bar{z}} = 8E_r$. The ribbons indicate the standard deviations $\Delta N_0/2$ and $\Delta\sigma_0/2$ of the central cloud.

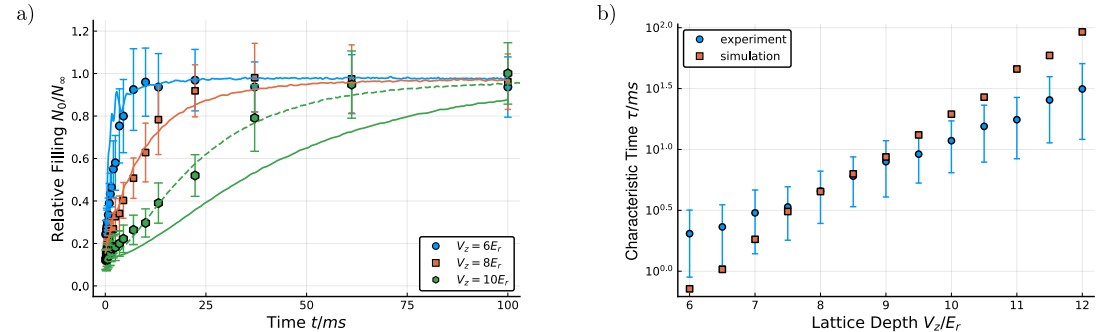

Figure 3: Refilling dynamics of the initially empty site. a) Comparison of the refilling dynamics of the simulation (solid lines) and the experiment. The additional dashed green line is equivalent to the solid green line but with the effective time $t_{\text{eff}} = 0.54\,t$ chosen to fit the experiment. b) Characteristic filling times $\tau$ of the simulation (squares) and experiment (circles) obtained by fitting the logistic function Eq. (35) to the data.

larger depths the simulations predict a refilling slower than the experiment. To quantify this, we have compared in Fig. 3b) the experimental and theoretical filling times $\tau$ in the logistic function, given above. One recognizes a good agreement for lattice depths on the order of $V_{\bar{z}} \lessapprox 10\,E_r$. A possible reason for the increasing discrepancy between the VTWA predictions and the experiment in deeper lattices is the simplistic Gaussian ansatz of the mode $\psi_{0,j}$, which is expected to break down at larger lattice depths.

In Fig. 4 we schematically present the refilling process in the picture of the harmonic oscillator eigenfunctions. The macroscopic occupation of the ground state induces an energy shift $\Delta E = U_{\text{eff}}\langle \hat{N}(\hat{N}-1)\rangle$ which causes particles to preferably tunnel into higher radial modes of the empty lattice site, leading to non-Gaussian wave function *tails*. Due to thermalization, the non-Gaussian transverse distribution condenses into the single-particle ground state but also causes the system to heat up. This intermediate step, which is ignored by our variational ansatz, becomes more pronounced as $U_{\text{eff}}/J$ or equivalently $V_{\bar{z}}$ increases. At even larger $V_{\bar{z}}$ the increasing effective interaction $U_{\text{eff}}$ further diminishes the quality of our theory as the validity of the TWA itself diminishes.

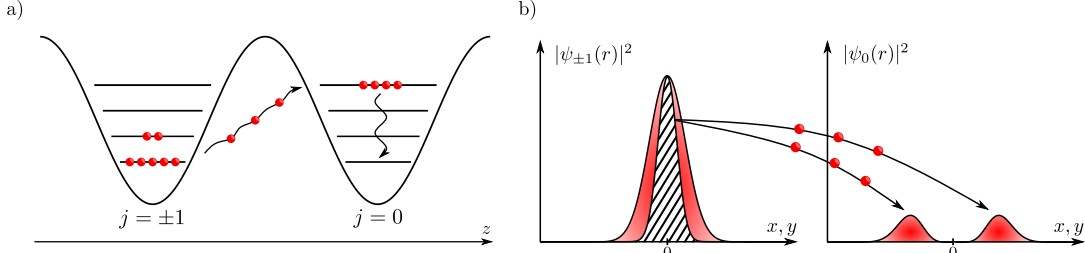

Figure 4: Schematic representation of the tunneling process between a full and an empty lattice site in terms of a) the single-particle eigenstates as well as b) the corresponding transversal wave functions. Due to the interaction shift a resonant transport between the single-particle ground states in neighboring sites with different occupation numbers is blocked and particles dominantly tunnel into higher radial modes of the initially empty site, which cannot accurately be expressed by our variational ansatz.

### 5.2.2 Number fluctuations

In contrast to the mean-field GPE approximation, the VTWA approach also allows us to calculate fluctuations of the occupation number. If one only considers the centralized second moments, diverging delta-function contributions from unoccupied modes cancel out:

$$
\begin{aligned}
(\Delta N_j)^2 &= \langle \hat{N}_j^2 \rangle - \langle \hat{N}_j \rangle^2 \\
&= \int dx\, dy \int d\tilde{x}\, d\tilde{y} \left\{ \overline{|\psi_j^{(0)}(x,y)\psi_j^{(0)}(\tilde{x},\tilde{y})|^2} - \overline{|\psi_j^{(0)}(x,y)|^2}\, \overline{|\psi_j^{(0)}(\tilde{x},\tilde{y})|^2} \right\} \\
&= \overline{N_j^2} - \overline{N_j}^2 .
\end{aligned}
\tag{36}
$$

The correlation function $g^{(2)}(0)$ of the central lattice site at different lattice depths is compared to experimental results in Fig. 5. The variational approach correctly predicts an initial rise of the number fluctuations. However, the amplitude of this peak deviates from the experimental data. At larger lattice depths, the positions of the maxima are also not correctly predicted. While the origin of the latter is the same as discussed above, the overestimation of the

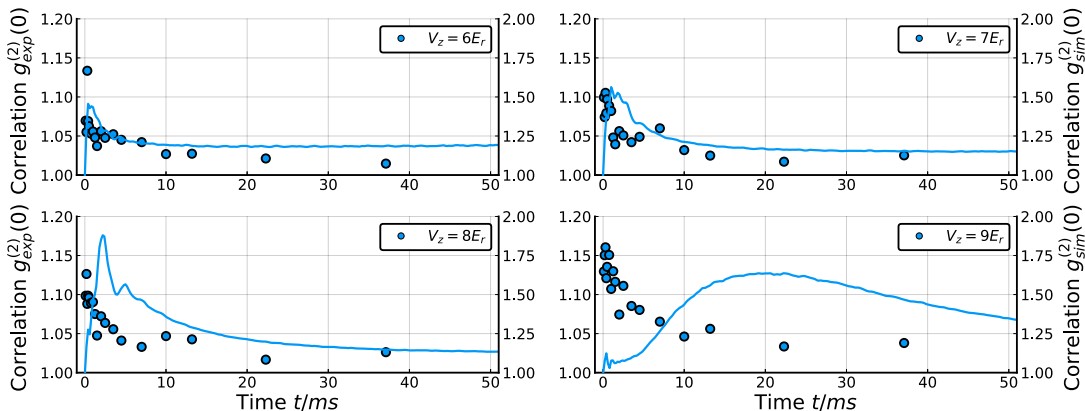

Figure 5: Correlation function $g^{(2)}(0)$ of the central atomic cloud for different lattice depths. Circles represent the experimental results and solid lines the simulations. Note the different scales of the correlation functions for simulation and experiment.

amplitudes of the $g^{(2)}$ correlations that are present at all lattice depths, is likely due to the effective few-mode description of the variational ansatz. The $g^{(2)}$ function of a system consisting of $M$ modes can be expressed in terms of number-correlations $\langle\langle\hat{n}_k\hat{n}_l\rangle\rangle = \langle(\hat{n}_k - \langle\hat{n}_k\rangle)(\hat{n}_l - \langle\hat{n}_l\rangle)\rangle$ between the modes

$$g^{(2)} = \frac{\langle\hat{N}(\hat{N}-1)\rangle}{\langle\hat{N}\rangle^2} = \frac{\displaystyle\sum_{l,m=1}^{M}\langle\hat{n}_l\hat{n}_m\rangle}{\langle\hat{N}\rangle^2} - \frac{1}{\langle\hat{N}\rangle} = 1 - \frac{1}{\langle\hat{N}\rangle} + \frac{\displaystyle\sum_{l,m=1}^{M}\langle\langle\hat{n}_k\hat{n}_l\rangle\rangle}{\langle\hat{N}\rangle^2}\,. \tag{37}$$

Typically every mode is correlated only with a finite number $d$ of other modes and thus the numerator of the last term scales with $dM$, while the denominator scales with $M^2$. As a consequence $g^{(2)}$ will be overestimated in simulations that do not take a sufficiently large number of effective modes into account. Increasing the number of effective modes, which corresponds to a larger number of stochastic variational parameter, does however increase the computational cost of the simulation and one has to find an optimal compromise.

### 5.2.3 Incoherent gains and losses

In the previous sections, only unitary dynamics have been considered. As a consequence of the choice of Hamiltonian, the diffusion matrix vanishes exactly. To demonstrate the inclusion of diffusion in the VTWA, we discuss additional incoherent driving and losses described by the Lindblad master equation on top of the previous Hamiltonian and variational ansatz:

$$\frac{d}{dt}\hat{\rho} = -\frac{i}{\hbar}[\hat{H},\hat{\rho}] + \frac{1}{2}\sum_{\mu}\int d\vec{r}\,(2\hat{L}_{\mu}\hat{\rho}\hat{L}_{\mu}^{\dagger} - \hat{L}_{\mu}^{\dagger}\hat{L}_{\mu}\hat{\rho} - \hat{\rho}\hat{L}_{\mu}^{\dagger}\hat{L}_{\mu})\,, \tag{38}$$

with incoherent gain rate $\Gamma_+$ and loss rate $\Gamma_-$

$$\hat{L}_+(\vec{r}) = \sqrt{\Gamma_+}\hat{\psi}^{\dagger}(\vec{r}), \quad \hat{L}_-(\vec{r}) = \sqrt{\Gamma_-}\hat{\psi}(\vec{r})\,. \tag{39}$$

These produce additional Wigner terms given by:

$$\frac{\partial}{\partial t}W|_{\Gamma_{\pm}} = -\left[\frac{\delta}{\delta\psi(\vec{r})}\frac{(\pm\Gamma_{\pm})}{2}\psi(\vec{r}) + \frac{\delta}{\delta\psi^*(\vec{r})}\frac{(\pm\Gamma_{\pm})}{2}\psi^*(\vec{r})\right]W$$
$$+ \left[\frac{\delta^2}{\delta\psi^*(\vec{r})\delta\psi(\vec{s})}\frac{\Gamma_{\pm}}{4} + \frac{\delta^2}{\delta\psi(\vec{r})\delta\psi^*(\vec{s})}\frac{\Gamma_{\pm}}{4}\right]W\,. \tag{40}$$

Using the change of variables given by eqs. (50) and calculating the second functional derivatives according to eq. (20), we can determine the corresponding diffusion matrix for the 4 variational parameters

$$D = \frac{\Gamma_{\pm}}{N}\begin{pmatrix} \frac{N^2}{2} & 0 & 0 & 0 \\ 0 & \frac{1}{4} & 0 & \frac{1}{8\sigma^2} \\ 0 & 0 & \frac{\sigma^2}{8} & 0 \\ 0 & \frac{1}{8\sigma^2} & 0 & \frac{1}{8\sigma^4} \end{pmatrix}\,, \tag{41}$$

which is positive. By also considering the drift term contributions, we find that the set of differential equations (30) receives additional (stochastic) terms:

$$dN_j|_{\Gamma_\pm} = \Gamma_\pm\left(\frac{1}{2} \pm N_j\right)dt + \sqrt{\Gamma_\pm N_j}\,dW_{N_j}, \tag{42a}$$

$$d\phi_j|_{\Gamma_\pm} = \sqrt{\frac{\Gamma_\pm}{2N_j}}\,dW_{\phi_j}, \tag{42b}$$

$$d\sigma_j|_{\Gamma_\pm} = \frac{3\Gamma_\pm\sigma_j}{8N_j}dt + \sqrt{\frac{\Gamma_\pm}{N_j}}\frac{\sigma_j}{2}\,dW_{\sigma_j}, \tag{42c}$$

$$dA_j|_{\Gamma_\pm} = \sqrt{\frac{\Gamma_\pm}{8N_j}}\frac{dW_{\phi_j} + dW_{A_j}}{\sigma_j^2}, \tag{42d}$$

i.e. are now SDEs with non-diagonal noise. The inclusion of noise is computationally slightly more taxing. More importantly, more trajectories are needed to obtain a good statistical convergence, i.e. have sufficiently smooth observables. Overall, the numerical integration of the set of SDEs is still efficient and feasible. Due to the specific choice of the variational ansatz, several terms contain $N_j^{-1}$ and the equations become unstable when $N_j \approx 0$.

To demonstrate the inclusion of dissipative effects, we consider a chain of 5 coupled condensates with incoherent gains and losses at each end respectively. As shown in the graphs 1a)

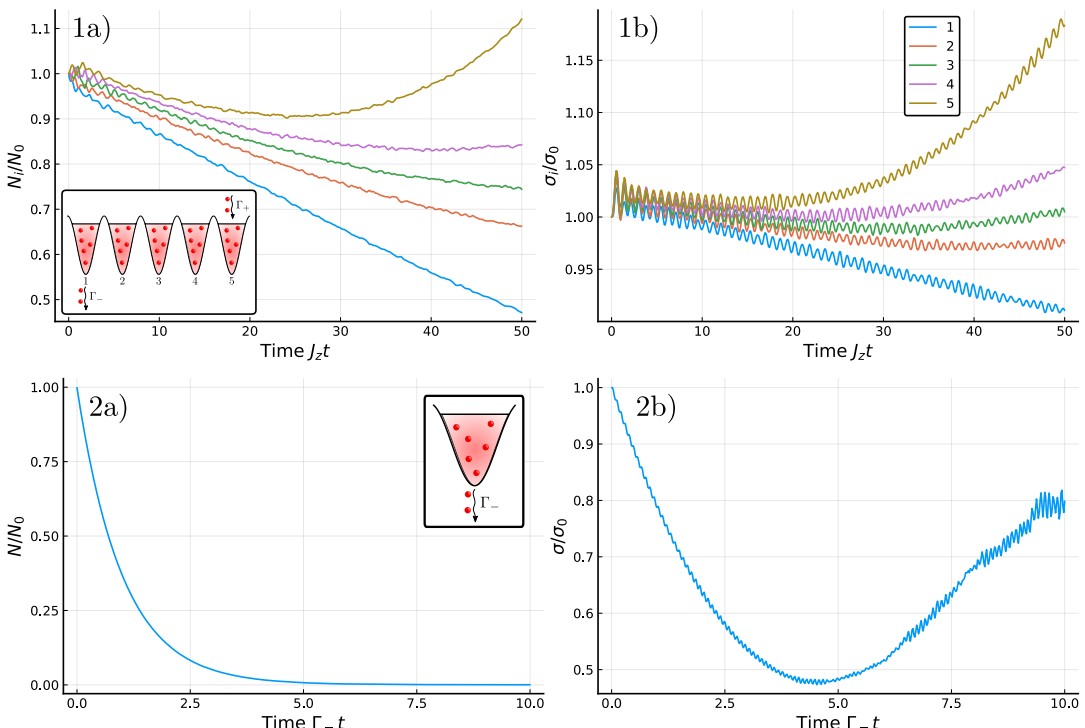

Figure 6: Condensate dynamics with incoherent gains and losses. 1a) Occupation numbers of 5 coupled condensates on a 1D lattice. The leftmost condensate is subjected to a loss rate of $\Gamma_-/J_z = 0.1$ and the rightmost to an incoherent gain of $\Gamma_+/J_z = 0.05$. All other parameters are taken from Sec. 5.1. 2a) Depletion of the occupation number of a single condensate with $\Gamma_-/V_r = 0.1$ and $\Gamma_+ = 0$. 1b) and 2b) show the respective transversal widths.

and 1b) of Fig. 6, the incoherent driving induces a particle difference between the two ends of the chain and thus a particle current. The radial widths follow the particle numbers of their respective site. In the graphs 2a) and 2b) the dynamics of a single condensate with incoherent losses are shown. As expected, the particle number decays exponentially with the rate $\Gamma_-$. The radial width decays as well, as it follows the particle number. When the condensate is almost depleted the overall width increases and the numerical noise in the radial width increases, eventually leading to a break-down of the method.

## 6   Summary and outlook

We derived a variational approach for the efficient simulation of weakly interacting Bose fields beyond the mean-field level using the truncated Wigner approximation and applied it to the filling dynamics of coupled Bose condensates in an optical lattice. This system is a paradigmatic example, where quantum fluctuations cannot be completely neglected, as the dynamics are strongly affected by modes with non-macroscopic occupation numbers. As a consequence, mean-field treatments such as the Gross-Pitaevskii equation give inaccurate results. At the same time, transverse degrees of freedom play an important role, as they affect the effective tunneling rates between neighboring lattice sites. This renders effective one-dimensional models inappropriate, which could in turn be treated fully quantum mechanically. The variational truncated Wigner approach is computationally inexpensive and gives access to dynamical properties such as average occupation numbers, transverse spatial profiles of the individual condensates in the lattice traps, as well as their respective fluctuations. We apply our findings to experimental observations of the out-of-equilibrium dynamics of a Bose gas in a one-dimensional optical lattice. We find quite good agreement with the experimental data, which is only limited by the adequacy of TWA and the quality of the variational ansatz. Therefore, we expect that the VTWA with an improved variational ansatz would yield results, which agree even better with the experimental data. Finally, the variational formulation can straightforwardly be extended to include, for instance, dissipation or thermal effects, which will be the subject of future studies.

## Acknowledgments

We thank Matthew Davis for discussions and acknowledge financial support by the Deutsche Forschungsgemeinschaft (DFG, German Research Foundation) via the Collaborative Research Center SFB/TR185 (Project No. 277625399).

## A   Mean-field model for the optical lattice

A classical approximation to the Hamiltonian (1) is given by the Lagrange function:

$$L_{\mathrm{cl}} = \int d\vec{r}\, \psi^*(\vec{r}) \left[ i\frac{\partial}{\partial t} + K\Delta - V_z \sin^2(\pi z) - V_r(x^2 + y^2) - \frac{U_0}{2}|\psi(\vec{r})|^2 \right] \psi(\vec{r}). \tag{43}$$

By inserting the variational ansatz Eq. (24) into the Lagrange function and applying the Euler-Lagrange equation [33]

$$\frac{\partial L_{\mathrm{cl}}}{\partial c} - \frac{d}{dt}\frac{\partial L_{\mathrm{cl}}}{\partial \dot{c}} = 0, \tag{44}$$

for each variational parameter $c \in \{N_j, \phi_j, \sigma_j, A_j \,|\, j \in \mathbb{Z}\}$, we obtain the same equations of motion as in Eqs. (30). However, in this classical case the initial conditions are deterministic. These equations have been used to generate the dashed curve in Fig. 1.

## B TWA and Bogoliubov approximation

In the following we will illustrate that the truncated Wigner approximation goes beyond a mean-field description of weakly interacting Bose gases and like Bogoliubov theory incorporates lowest-order quantum fluctuations [15]. To this end we consider the Bose-Hubbard model (3) as a generic example. Expanding the quantum "fields" $\hat{a}_j = \alpha_j + \hat{b}_j$ around the mean-field solution $\alpha_j$ of

$$\frac{d}{dt}\alpha_j = -i\omega\alpha_j - iU|\alpha_j|^2\alpha_j + iJ(\alpha_{j-1} + \alpha_{j+1}), \tag{45}$$

and neglecting higher than second order terms in the quantum fluctuations yields the Bogoliubov approximation to the Bose-Hubbard Hamiltonian

$$
\begin{aligned}
\hat{H}_{\mathrm{BH}} &\approx \mathcal{H}_{\mathrm{BH}}^{\mathrm{MF}} + \hbar\omega\sum_j(\alpha_j^*\hat{b}_j + h.a.) + \hbar\omega\sum_j\hat{b}_j^\dagger\hat{b}_j + \hbar U\sum_j|\alpha_j|^2(\alpha_j^*\hat{b}_j + \alpha_j\hat{b}_j^\dagger) \\
&\quad + \hbar U\sum_j\left(2|\alpha_j|^2\hat{b}_j^\dagger\hat{b}_j + \frac{1}{2}(\alpha_j^2\hat{b}_j^{\dagger 2} + \alpha_j^{*2}\hat{b}_j^2)\right) - \hbar J\sum_{\langle jk\rangle}\left(\alpha_j^*\hat{b}_k + \mathrm{h.a.}\right) - \hbar J\sum_{\langle jk\rangle}\hat{b}_j^\dagger\hat{b}_k,
\end{aligned}
\tag{46}
$$

where $\mathcal{H}_{\mathrm{BH}}^{\mathrm{MF}}$ denotes the mean-field Hamiltonian which is a c-number. The equation of motion of the Wigner distribution $W(\beta_j, \beta_j^*)$ of quantum fluctuations $\hat{b}_j$ and $\hat{b}_j^\dagger$ can straightforwardly be obtained from the transformation rules (6). Despite the presence of quadratic terms $\sim \hat{b}_j^2$ and $\hat{b}_j^{\dagger 2}$ this yields a FPE with first-order derivatives only and truncated and full Wigner distributions are identical. Thus, the TWA provides exact results for the dynamics under the Bogoliubov Hamiltonian (46).

## C Fokker-Planck equations and change of variables

The Fokker-Planck equation is a partial differential equation governing the time evolution of a (quasi-) probability distribution $W$:

$$\frac{\partial}{\partial t}W = \mathcal{L}W, \tag{47}$$

where the choice of the operator $\mathcal{L}$ determines the type of the FPE. The *multivariate* FPE is given by:

$$\mathcal{L}(\vec{x}) = -\sum_i\frac{\partial}{\partial x_i}D_{x_i} + \sum_{ij}\frac{\partial^2}{\partial x_i\partial x_j}D_{x_i,x_j}, \tag{48}$$

where $\vec{x}$ is a finite set of continuous variables. In the continuum limit, where there is a complex field $\psi(\vec{r}), \psi^*(\vec{r})$ for every $\vec{r} \in \mathbb{R}^n$, one obtains a *functional* FPE:

$$
\begin{aligned}
\mathcal{L}[\psi, \psi^*] = &-\int d\vec{r}\,\frac{\delta}{\delta\psi(\vec{r})}D_{\psi(\vec{r})} \\
&+ \int d\vec{r}\,d\vec{s}\left(\frac{\delta^2}{\delta\psi(\vec{r})\delta\psi(\vec{s})}D_{\psi(\vec{r}),\psi(\vec{s})} + \frac{\delta^2}{\delta\psi^*(\vec{r})\delta\psi(\vec{s})}D_{\psi^*(\vec{r}),\psi(\vec{s})}\right) + \mathrm{c.c.}.
\end{aligned}
\tag{49}
$$

The coefficients $D_.$ form the so called drift vector and the coefficients $D_{..}$ with two indices the diffusion matrix. Only a PDE with a positive-semidefinite diffusion matrix is considered a FPE and has an equivalent stochastic differential equation [40, 41].

Assume a multivariate FPE in $\vec{x} \in \mathbb{R}^n$ and let $\vec{y} = \vec{y}(\vec{x}) \in \mathbb{R}^n$ be a new set of variables which is expressed as a function of the old ones. Then an equivalent FPE in the new variables exists [41]. Similarly, if we assume a functional FPE for $W[\psi, \psi^*]$, a new set of variables $\vec{c} \in C$ can be expressed as functionals of the field $\psi(\vec{r}), \psi^*(\vec{r})$. The new drift and diffusion coefficients are obtained from taking the continuum limit of the multivariate transform:

$$D_{c_i} = \int d\vec{r} \frac{\delta c_i}{\delta \psi(\vec{r})} D_{\psi(\vec{r})} + \int d\vec{r} d\vec{s} \frac{\delta^2 c_i}{\delta \psi(\vec{r}) \delta \psi(\vec{s})} D_{\psi(\vec{r})\psi(\vec{s})}$$
$$+ \int d\vec{r} d\vec{s} \frac{\delta^2 c_i}{\delta \psi^*(\vec{r}) \delta \psi(\vec{s})} D_{\psi^*(\vec{r})\psi(\vec{s})} + \text{c.c.} \,, \tag{50a}$$

$$D_{c_i,c_j} = \int d\vec{r} d\vec{s} \left( \frac{\delta c_i}{\delta \psi(\vec{r})} \frac{\delta c_j}{\delta \psi(\vec{s})} D_{\psi(\vec{r}),\psi(\vec{s})} + \frac{\delta c_i}{\delta \psi^*(\vec{r})} \frac{\delta c_j}{\delta \psi(\vec{s})} D_{\psi^*(\vec{r}),\psi(\vec{s})} \right) + \text{c.c.} \,. \tag{50b}$$

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
