# Peer review of "Variational truncated Wigner approximation for weakly interacting Bose fields: Dynamics of coupled condensates"

_SciPost Physics, doi:SciPost Phys. 12, 051 (2022)_

## Round 1 · Referee Report · Anonymous (Referee 1) · 2021-8-20

Strengths

1- This paper contains one of the clearest derivations and presentations of the Truncated Wigner method

2- There is an amazing numerics vs. experiment correspondence, achieved with a modest numerical effort

Weaknesses

1- Throughout the paper, the source of the experimental data is not clear. It is not even clear is the data comes form a previous work or it is original. The answer to this question will affect my opinion on what to do with the manuscript in the second round.

2- If the experimental data is not original, the rest is a standard truncated Wigner with an additional variational simplification for the mean-field propagations. I do not think it is a significant theoretical advancement. On the other hand, if the paper is combination of a numerics and an experiment, the synergy of it would serve as a powerful inspiration for future projects of that sort.

One of the authors of the experimental paper [4] is on the author list: does this mean that the data of [4] was reused in this manuscript? Even in this case, the paper may be publishable, in my opinion.

3- Smaller remarks :

(i) Appearance of the "residual field" in the abstract is sudden, and its meaning
is unclear at that stage. In fact, it remains unclear all the way till the middle of
page 4;

(ii) At Fig. 1, it is not clear what the number of atoms, transverse trapping
frequency, lattice depth, and the coupling constants are. It may be implied that
these parameters are the same as presented at page 5, but it is not clear is this is
the case. All other plots also suffer from incomplete captions;

(iii) P. 3, second column: The phrase "...where for all
practical purposes N < 1 must be chosen" is cryptic, I suspect a misprint.

Report

As I said above, my decision on whether the manuscript should be accepted
depends on whether it is a numerics-experiment collaboration or pure
numerics. I would reject the paper if the latter is the case.

Requested changes

I've listed the suggested changes in the "Weaknesses" section.

  • validity: top
  • significance: low
  • originality: ok
  • clarity: top
  • formatting: perfect
  • grammar: perfect

Author:  Christopher Mink  on 2021-11-29  [id 1988]

(in reply to Report 1 on 2021-08-20)

We thank Referee 1 for the strong appreciation of our work.

Regarding the weaknesses: 1) The experimental data in the manuscript do not stem from a previous work but are measured anew. The motivation to revisit this kind of experiment is that previously only average occupation numbers were measured and published. These measurements have now been refined to extract the characteristic time scales of the refilling as a function of the lattice depth quantitatively. Furthermore, fluctuations of the occupation numbers have been measured and are presented in the paper. In order to clarify this point we have expanded the description of the experimental set up in Section 5.1.

2) We respectfully disagree with the statement of the referee that there is no significant theoretical advancement. Our method is not the standard truncated Wigner approach. The standard TWA for continuous systems requires the decomposition into a set of modes of static eigenfunctions whose number is then truncated based on physical assumptions. To cover the physics of interacting condensates, the number of modes that need to be taken into account is not small. As a consequence, evolving the individual trajectories is computationally expensive. For example, in simulations done in the group of Matthew Davis (private communication) on a similar problem of up to 21 coupled 2D condensates, a single trajectory on a desktop computer requires a calculation time of roughly 1 hour. In order to run O(100) trajectories, a super computer is strictly required. On the other hand, using our variational ansatz - which reduces the dynamics to 4 degrees of freedom per lattice site - the simulation of a single trajectory takes only 1 ms. So the main achievement of the variational ansatz is a substantially reduced numerical effort and thus gives access to larger and potentially more complex systems. However, as also explained in the response to referee 2, in the nonlinear mapping of the full TWA problem (including dissipation and noise) to a variational ansatz, it is not clear up-front that the resulting stochastic equations for the variational parameters behave properly, i.e. have positive definite diffusion. As outlined in the new sub-section 5.2.3 where gain and loss processes were included, this is however the case. Secondly, indeed, the paper represents a combined theory-experiment case study with new experimental data (see reply to comment 1). It was our original motivation to provide a powerful inspiration for future projects of that sort. We are glad that Referee 1 recognizes our original intention.

Concerning the smaller remakrs: (i) We agree with the referee. We follow the suggestion of the referee and mention the "residual field" first in the introduction. In this way we clarify our variational procedure right from the beginning. (ii) We apologize for this lack of clarity. We now precisely state all relevant quantities in the captions of all figures in order to guarantee their reproducibility. (iii) Indeed, in the revised manuscript we corrected that statement.

---

## Round 1 · Referee Report · Anonymous (Referee 2) · 2021-8-31

Strengths

-) Development of a potentially very powerful numerical approach to simulate the dynamics of BECs and related multi-mode bosonic systems.

-) Direct comparison between experiment and theory.

Weaknesses

-) Based on the discussed example, the broader range of applicability and validity of the method is not clear.

Report

In this paper the authors introduce a variational truncated Wigner approximation (TWA) scheme to simulate the dynamics of a multi-mode bosonic system. The key idea is to use a variational ansatz for the many-body wavefunction and then to derive an effective stochastic equation for the variational parameters. This allows one to simulate much larger systems than what would be possible by using a standard TWA approach. The method is then applied for the simulation of the refilling dynamics of an empty site in quasi 1D optical lattice. This is a problem where other methods such as mean-field theory or tDMRG simulations of the 1D Bose Hubbard model become very inaccurate. The numerical results are benchmarked in detail by actual experimental measurements. Overall there is a good agreement between the numerical and the experimental results for the mean occupation numbers, but a significant difference still is observed for the number fluctuations. The authors explain this discrepancy by the too simplistic variational ansatz used in their simulations and conjecture that this aspect can be considerably improved in more extensive calculations.

The numerical simulation method described in this paper seems to be very powerful and the detailed derivation presented in this manuscript will be useful for generalizing this scheme for many applications in the field of cold atoms and quantum optics. However, based on the illustrated example this is more of a hope and it is not yet clear that the method can indeed be applied for a more general set of problems:

1) In the current example there are no diffusion terms included. Thus the method reduces to the deterministic equations discussed in Appendix A (and derived in previous works) with random initial conditions. Including diffusion terms will not only increase the computation time, but may introduce additional numerical artefacts, such diverging trajectories in positive-P function simulations. Further, is it clear that the resulting diffusion terms will remain positive under the transformation from fields to variational parameters? Especially this last point is very important and should be discussed.

2) The example discussed in the paper shows that the method capture rather poorly the fluctuations in the atom number. The authors claim that this can be improved by improving the variational ansatz. But this is not shown. More precisely, the question is if the accuracy in the number fluctuations can be improved without losing the numerical benefit from the reduced set of variational parameters.

In summary, this is a very interesting idea and in general also a well presented paper. The direct comparison between simulations and experiments shows that the proposed method is capable of making experimentally relevant predictions for this setting, which are very hard to obtain otherwise. However, the same comparison also shows that fluctuations are captured very poorly by the simulation and it still needs to be shown that this aspect can be improved in a systematic and practical manner. Therefore, in its current form I don't think that the paper meets the strict acceptance criteria for a publication in SciPost Physics.

Minor comments:
-) In Figure 1, please add the the total number of atoms used in this simulation. This is relevant to understand in which parameter regime the different methods are compared.

-) In Eq. (21) the field \Psi_1 is factored out and also below Eq. (28) it says that the dynamics of \psi_0 and \psi_1 are completely decoupled. But without going through the derivation it is not fully clear if \psi_1 is only adiabatically eliminated or does not affect the dynamics of \psi_0 at all. In the general, at which point do the vacuum fluctuations enter the results? Only through initial conditions? I think this is an important point, which is not immediately apparent from the current discussion.

Requested changes

-) Please add a discussion or even a simple illustrative example about the positivity of the diffusion matrix for the variational parameters.

-) Please demonstrate the ability to improve the calculation of number fluctuations by an improved variational ansatz. This could be done, for example, for a simpler two-site model.

  • validity: high
  • significance: good
  • originality: good
  • clarity: high
  • formatting: excellent
  • grammar: perfect

Author:  Christopher Mink  on 2021-11-29  [id 1987]

(in reply to Report 2 on 2021-08-31)

We thank Referee 2 for a critical reading of our manuscript and for the critical comments, which aim at improving it further.

1) In the previous version of the manuscript, only unitary dynamics have been considered. As a consequence of the Hamiltonian in question, the diffusion matrix vanishes exactly. In order to demonstrate the inclusion of diffusion in the VTWA, we have added a new Subsection 5.2.3. There we discuss additional incoherent driving and losses described by the Lindblad master equation on top of the previous Hamiltonian. Most importantly, we demonstrate that the resulting stochastic differential equations of the variational parameters are still well behaved, i.e. result from a positive diffusion matrix of the corresponding Fokker-Planck equation.

2) We agree with the referee that the quantitative agreement for the fluctuations is lacking. We do want to point out though, that the most significant feature, namely a local maximum of the number fluctuations in the filling dynamics is captured by the numerics. As mentioned in the reply to referee 1, the reduction of the full problem to four effective degrees of freedom per site in the variational ansatz substantially reduces the numerical effort. As is now explained in the revised text of sub-section 5.2.2, this truncation of the number of modes unfortunately also leads to overestimating super-Poissonian number fluctuations. Thus one has to find a suitable compromise between keeping the numerics computationally inexpensive while still correctly accounting for fluctuations. For the present discussion, which was intended as a joint experiment-theory proof-of-principle of the variational method, an extension to a more complex variational ansatz was too resource intensive.

We appreciate the general positive evaluation of our manuscript by Referee 2. Furthermore, we think that we have properly taken the fundamental suggestions of Referee 2 into account by expanding the manuscript correspondingly.

Regarding the minor comments: - The total number of atoms used in the simulations in now mentioned in Fig. 1. - We agree that we have been a bit unclear about the treatment of the residual field. As is now stated more clearly in the text, the backaction of $\psi_1$ to the dynamics of $\psi_0$ is neglected. I.e. vacuum fluctuations contained in the residual field do not affect the evolution of the macroscopic field $\psi_0$. However, part of the quantum noise is still accounted for via the non-deterministic initial conditions of the variational parameters. This is the central improvement over the deterministic Gross-Pitaevskii equation that remedies the abrupt changes of the particle number, leading to results that are in good agreement with the experimental data.

---

## Round 2 · Referee Report · Anonymous (Referee 1) · 2021-11-29

Report

I am satisfied with the corrections and answers to my questions, I recommend publishing.

---

## Round 2 · Referee Report · Anonymous (Referee 2) · 2021-12-1

Report

In the revised version of the paper and in their reply, the authors address most of the point that I raised in my previous report. In particular, a whole new section demonstrating the simulation of a dissipative system has been added, which nicely illustrates the generality of the described method. Overall, I am satisfied with the reply by the authors and the changes in revisions in the manuscript. Since the method seems to be very powerful and applicable for a broad range of problems, I recommend to publish the current version of the paper in SciPost Physics.

---

## Round 2 · Author Response

We are grateful for the constructive reports from both referees and for the opportunity to carefully revise the manuscript accordingly. The text has been significantly improved based on the respective suggestions.

---

## Round 2 · List of Changes

-The first mention of the residual (non-macroscopic) field has been moved to the introduction
-the caption of figure 1 now includes all relevant parameters.
-The change of variables in the functional Fokker-Planck equation and subsequent neglection of the residual field on p. 8-9 (most importantly eq. 22) have been overhauled to make the decoupling of the fields $\psi_0$ and $\psi_1$ more clear.
-a newly added paragraph at the end of section 5.1 motivates the experimental data and stresses that it is - in fact - newly obtained data.
-Section 5.2.2 has been reworked to explain why the overestimation of the number fluctuations is a generic artifact of few-mode approximations such as our variational ansatz.
-The new section 5.2.3 introduces local incoherent gains and losses to the optical lattice. We derive the positive-semidefinite diffusion matrix within the variational scheme and derive a set of stochastic differential equations. The incoherent contribution to the dynamics for a coupled and a single trap is shown in the new figure 6.

---

## Editorial Decision

published